# Dysnatremia as a Mortality Marker in Intensive Care Patients with SARS-CoV-2 Infection: A Retrospective Study

**DOI:** 10.3390/medicina60071019

**Published:** 2024-06-21

**Authors:** Guler Eraslan Doganay, Melek Doganci, Gulsah Yurtseven, Azra Ozanbarci, Abdullah Kahraman, Mustafa Ozgur Cirik, Fatma Ozturk Yalcin, Seray Hazer, Kerem Ensarioglu

**Affiliations:** 1Department of Anesthesiology and Reanimation, Ankara Ataturk Sanatorium Training and Research Hospital, University of Health Sciences, 06290 Ankara, Turkey; melekdidik@hotmail.com (M.D.); gulsahyurtseven@yandex.com (G.Y.); dr.ozgurr@hotmail.com (M.O.C.); ftmozt@yahoo.co.uk (F.O.Y.); 2Anesthesiology and Reanimation Intensive Care Unit, Ministry of Health Ankara Etlik City Hospital, 06170 Ankara, Turkey; azraarslan@yahoo.de (A.O.); abdullahhero100@gmail.com (A.K.); 3Department of Thorasic Surgery, Ankara Ataturk Sanatorium Training and Research Hospital, University of Health Sciences, 06290 Ankara, Turkey; drserayhazer@gmail.com; 4Department of Pulmonology, Ankara Ataturk Sanatorium Training and Research Hospital, University of Health Sciences, 06290 Ankara, Turkey; kerem.ensarioglu@gmail.com

**Keywords:** COVID-19, dysnatremia, hypernatremia, hyponatremia, mortality, SARS-CoV-2

## Abstract

*Background and Objectives*: The Severe Acute Respiratory Syndrome Coronavirus 2 (SARS-CoV-2) infection may cause acute respiratory failure, but also remains responsible for many other pathologies, including electrolyte disorders. SARS-CoV-2 infection causes disorders in many systems and can disrupt water homeostasis with thirst and appetite abnormalities. Dysnatremia affects prognosis, and may be associated with mortality in patients admitted to an intensive care unit (ICU) diagnosed with SARS-CoV-2. *Materials and Methods*: The study included 209 patients admitted to the ICU between 12 April 2021 and 1 March 2022 who were over 18 years old and diagnosed with SARS-CoV-2 infection by clinical and thoracic tomography findings or with a positive reverse transcription polymerase chain reaction (RT-PCR) test result. The laboratory markers, treatment modalities, nutritional, and respiratory support also for outcome evaluation, length of stay in the ICU, total hospitalization duration, and mortality in the ICU were recorded. The laboratory marker comparison was made using admission with the final assessment performed before the time of mortality in the ICU or after discharge. *Results*: Inotropic requirements among patients were high, which reflected mortality in the ICU. Hypernatremia presence was associated with an increase in enteral support, the inotropic support requirement, and mortality. Hypernatremia was correlated with diabetes mellitus, chronic renal failure, and a longer duration under mechanical ventilation. *Conclusions*: Hypernatremia was an important risk factor in ICU patients hospitalized for SARS-CoV-2 infection, which was also affected by the treatment regimens given themselves. This complex relationship underlies the importance of proper electrolyte management, especially in patients who were under severe stress and organ failure.

## 1. Introduction

The outcome of SARS-CoV-2 infection in patients is highly variable. The World Health Organization (WHO) reported approximately 5,584,374 deaths due to this disease in January 2022, nearly two years after the start of the SARS-CoV-2 pandemic [1]. A wide spectrum of prognoses is observed in patients. Starting from tissue and organ damage, lung, heart, and kidney failure may occur, leading to multiple systemic inflammatory reactions in some patients [2]. 

Although most cases diagnosed as SARS-CoV-2 were mild, acute respiratory failure, heart disease, and kidney failure were accompanying diseases, especially in cases that resulted in death. This situation once again emphasized the importance of prognostic markers in terms of the early recognition of patients at high risk of both morbidity and mortality, and in terms of the efficient use of healthcare resources in cases requiring hospitalization [3]. SARS-CoV-2 infection is also responsible for many other pathologies, including electrolyte disturbances [4,5,6].

Changes in plasma sodium concentration are determined by changes in water balance, independent of the total body sodium amount. The two basic mechanisms responsible for regulating water metabolism are the antidiuretic hormone and the feeling of thirst [7]. While SARS-CoV-2 infection can cause multiple organ failure, it has also been reported to cause dysfunction of the renin-angiotensin-aldosterone system, disrupting water homeostasis with thirst and appetite abnormalities. Hyponatremia occurs due to the effect of antidiuretic hormone (ADH). Symptoms are typically dependent on the rapidity of onset of the clinical condition. Neurological symptoms are possible in severe hyponatremia (serum sodium < 120 mEq/L). Hypernatremia is usually caused by either a deficit of total body water or by an inappropriately high sodium input. In general, however, even during infusion of large amounts of sodium-containing solutions (as during treatment of acute hypovolemia), hypernatremia is infrequently observed and less pronounced.

Various electrolyte disorders were observed in SARS-CoV-2 patients, but hypo-hypernatremia was particularly striking [8]. Although hyponatremia is more common in SARS-CoV-2-infected patients, the presence of hypernatremia was associated with higher mortality and a requirement for MV [9]. Many studies have focused on hyponatremia, which occurs commonly in SARS-CoV-2 cases and may be a marker of disease severity [10,11,12,13]. It has been stated that the most important risk factors for death in patients presenting with hyponatremia are hypoxia and sepsis [14]. 

Few studies have examined dysnatremia as a predictor of mortality in patients with SARS-CoV-2 infection [3,15]. HOPE is an international study registry of 4664 hospitalized patients with SARS-CoV-2 infection. In this study, 20.5% of patients were reported to have hyponatremia, and hypernatremia was observed in 3.7% of patients [16]. It has been reported in many publications that hyponatremia may be due not only to hypovolemia, but also to some hypothalamic arginine vasopressin secretion stimulation [17,18,19,20].

Serious-life-threatening hypernatremia has also been observed during hospital admission in SARS-CoV-2 cases [21,22].

Therefore, it can be predicted that dysnatremia affects prognosis and may be associated with mortality in patients admitted to intensive care with SARS-CoV-2 infection.

The aim of this study is to determine the relationship of dysnatremia with prognosis and mortality in SARS-CoV-2 intensive care patients. Patients were classified as hyponatremic (Blood sodium level < 135 mmol/L), eunatremic (Blood sodium level 135–145 mmol/L), and hypernatremic (Blood sodium level > 145 mmol/L) on admission to the intensive care unit (ICU).

## 2. Patients and Method

After the study protocol was approved by Ankara Ataturk Sanatorium Training and Research Hospital Ethics Committee (Ethical Decision No: E-53610172-799-206667331 dated 11.01.2023), patients whose admissions were made between 12 April 2021 and 1 March 2022 were retrospectively scanned from the hospital database and patient files. The time period was chosen because the intensive care unit was solely assigned to COVID-19 patients during the mentioned period.

The study included patients admitted to the ICU who were over 18 years old and were diagnosed with SARS-CoV-2 infection by clinical and thoracic tomography findings or with a positive reverse transcription polymerase chain reaction (RT-PCR) test result. 

Demographic data of the patients, body mass index, additional comorbidities if present, and laboratory sampling results at the time of admission were the main evaluated data. The laboratory markers included routine blood count, liver and renal function tests, inflammatory markers, and routine testing performed for COVID-19, which included ferritin, procalcitonin, LDH, and d-dimer levels. Treatment modalities in the ICU were also recorded, consisting of nutritional support requirements and types, COVID-19 treatment regimens, inotropic support, glucocorticoid treatment regimens, and respiratory support requirements. For outcome evaluation, length of stay ICU (LOS ICU), total hospitalization duration, and mortality in ICU were recorded.

The laboratory marker comparison was made with the final assessment performed before the time of exit or discharge from the ICU. 

Patients under the age of 18, who had PCR (-), and whose clinical and thoracic tomography findings were not suggestive of SARS-CoV-2 infection were excluded from the study. Those who had been re-evaluated at the ICU for other possible diagnoses and were later transferred to other ICUs, such as those admitted with clinical suspicion, were also removed from the study.

### Statistical Evaluation

The patients’ results were put into a Microsoft Excel Windows 365 MSO (version 2405) file for overall evaluation. After investigating any incorrectly entered values, the data were moved to a statistics module (IBM Corp. Released 2017. IBM SPSS Statistics for Windows, Version 25.0). The initial assessment was performed using descriptive analysis, for which values were given with mean and standard deviation or with median and percentiles as required. Parametric distribution was evaluated using a Q-Q plot analysis. Paired sample *t*-test comparisons were made between groups for parametric values. Correlation analyses were performed using Spearman’s correlation. Binomial regression analysis was performed to evaluate the role of any parameters as an independent factor. *p* values at or below 0.05 were accepted as statistically significant.

## 3. Results

A total of 209 patients were included in the study. The majority of the patients were male (*n* = 114, 54.5%). The average age of the patients was 68.1 (±13.8) years, and the body mass index (BMI) was found to be 26.6 (±2.6). A median of 5 (2–10) days was observed between RT-PCR positivity and hospital admission. The median duration of total hospitalization, LOS ICU, days on invasive mechanical ventilation (IMV), and non-invasive mechanical ventilation (NIMV) were reported to be 11 (5–18), 6 (2–12), 1 (0–6), and 2 (1–5) days, respectively. In total, 78 patients (37.3%) showed culture positivity regarding additional bacterial involvement. In the culture of patients, the predominant organism was *Staphylococcus aureus* (*n*:31), followed by Candida (both albicans and non-albicans) (*n*:24), *Enterococcus*, *Escherichia coli*, *Acinetobacter*, *Klebsiella pneumonia*, and *Pseudomonas aeruginosa*. In some patients, there was also polymicrobial growth present. Diabetes mellitus (*n* = 66, 31.6%) and hypertension (*n* = 86, 41.1%) were the most observed comorbidities. Regarding nutritional support requirements, 20 (9.6%) patients required total parenteral support, while nearly half of the patients (*n* = 110, 52.6%) had enteral support requirements.

Favipiravir (*n* = 102, 48.8%) and intravenous glucocorticoid regimens were the mainstay of the treatment administered in the ICU. Inotropic requirements among patients were high (*n* = 124, 59.3%), which was reflected in the ICU mortality (*n* = 124, 59.3%) (Table 1).

The comparison of laboratory parameters made at admission and last evaluation showed that, in routine blood counts, hemoglobin, white blood cell, lymphocyte, and platelets had changed significantly (13.03 g/dL to 12.05 g/dL, *p* = 0.001; 13.33 × 10^9^/L to 15.06 × 10^9^/L, *p* = 0.002; 0.96 × 10^9^/L to 1.25 × 10^9^/L, *p* = 0.001, and 258.23 × 10^9^/L to 218.51 × 10^9^/L, *p* = 0.001, respectively). A treatment response favoring reduced inflammatory markers was also observed in the C-reactive protein and Erythrocyte Sedimentation Rate (ESR) evaluation (127.22 mg/L to 92.3 mg/L, *p* = 0.001 and 60.44 to 50.47, *p* = 0.001, respectively). Ferritin, d-dimer, and procalcitonin levels did not show significant change. 

Lactate dehydrogenase (LDH), creatinine, aspartate aminotransferase (AST), and alanine transaminase (ALT) were other parameters found to be statistically different (612.72 U/L to 877.96 U/L, *p* = 0.01; 1.17 mg/dL to 1.43 mg/dL, *p* = 0.001; 141 U/L to 308 U/L, *p* = 0.041; and 109 U/L to 198 U/L, *p* = 0.038, respectively). Sodium value also varied from the initial admission result to the last evaluation, with a mean of 139 mEq/L to 142 mEq/L and a *p*-score of 0.001 (Table 2). 

Correlation analyses were made with sodium levels being divided into three categories: hyponatremia, normal, and hypernatremia. Hypernatremia was correlated with diabetes mellitus, chronic renal failure, and a longer duration under mechanical ventilation (r(209) = 0.144, *p* = 0.037; r(209) = 0.184, *p* = 0.008; r(209) = 0.171, *p* = 0.014, respectively). Hypertension and chronic renal failure were also negatively correlated with normal blood sodium levels (r(209) = −0.139, *p* = 0.044, and r(209) = −0.168, *p* = 0.015). For treatment correlation, the presence of hypernatremia was associated with an increase in enteral support and inotropic support requirements and mortality (r(209) = 0.242, *p* = 0.001 and r(209) = 0.161, *p* = 0.001). Hyponatremia did not have any correlation with comorbidities and treatment modalities (Table 3). 

Regression analyses were performed for the role of hypernatremia, with three models for mortality, enteral support, and inotropic support. All models had a value higher than 0.5 for the Hosmer and Lemeshow test (0.106, 0.085, and 0.053, respectively). The models’ Nagelkerke R square results were 0.706, 0.870, and 0.666, with each model correctly classifying at least 60% of the given data. Hypernatremia’s role in mortality in patients in the ICU was not statistically significant in the regression analyses (*p* values of 0.339 and 0.417, respectively) (Table 4). Hypernatremia’s role in inotropic support was not statistically significant in the regression analyses (*p* values of 0.339 and 0.417, respectively). However, the regression analysis was significant when hypernatremia and enteral support were evaluated (*p*: 0.031). 

## 4. Discussion

This study includes an adequate number of patients with varying requirements for Non-Invasive Mechanical Ventilation (NIMV) and Invasive Mechanical Ventilation (IMV). Comorbidities were within the expected ranges for the patients’ age groups. The increased durations of invasive mechanical ventilation, requirement for inotropic support, and mortality in the ICU support the assumption of a severe condition.

This study demonstrated the association of hypernatremia in SARS-CoV-2-infected patients in the ICU with an IMV, requirement for inotropic support, and enteral nutrition. Another finding of the study, the association of hypernatremia with comorbidities such as diabetes mellitus and renal failure, suggests that these conditions exacerbate the disease course. 

Changes in routine blood count, favoring reduced inflammatory markers, indicate an overall treatment response, which was also reflected in CRP and sedimentation levels. However, impaired renal and liver function tests suggest organ failure and severity of the disease in patients. Changes in sodium levels were affected by enteral nutrition, which may be associated with enteral feeding during the invasive mechanical ventilation process in critically ill patients.

In critically ill patients without COVID-19, multiple studies have demonstrated that hypernatremia is a significant predictor of mortality [23,24]. Hypernatremia may be linked to hyperosmolar oxidative cell stress through inflammation, which may contribute to mortality [25].

The relationship between SARS-CoV-2 infection and hypernatremia has not been well understood, but it has been attributed to dehydration secondary to low oral intake, insensible fluid losses from fever, and dehydration associated with gastrointestinal symptoms. The relationship between SARS-CoV-2 infection and hyponatremia has been postulated to be a syndrome of inappropriate antidiuretic hormone secretion and inflammation [15].

Regarding the evaluation of sodium levels, the association between hypernatremia and prolonged invasive mechanical ventilation duration and renal failure may also be related to the increased duration of enteral nutrition support. Similarly, there was an association between the requirement for inotropic support and mortality, indicating that patients with more severe conditions, requiring additional respiratory and cardiac support, are prone to hypernatremia. However, regression analysis showed that hypernatremia was not significant in terms of mortality and requirement for inotropic support, but maintained its significance in terms of enteral support. Although we predicted dysnatremia in our study, our results did not observe hyponatremia as a significant risk factor.

Aggarwal et al., in a study conducted in the USA, reported that 50% of patients had hyponatremia [26]. Similarly, in the HOPE study, hyponatremia was observed in 20.5% of patients, while hypernatremia was seen in 3.7%, with both conditions being reported as independent risk factors for mortality and sepsis in patients hospitalized with SARS-CoV-2 pneumonia [16]. Unlike the HOPE study, hypernatremia was prominent as a risk factor in our study, while hyponatremia was not found to be a significant risk factor. Zimmel and colleagues reached similar findings to our study, with an association found between hypernatremia and prolonged MV duration and overall ICU duration [27].

In the HOPE study conducted by Ruiz-Sanchez et al., patients admitted to the ICU with pneumonia had an overall disposition to hyponatremia compared to hypernatremia upon admission. This was attributed to the syndrome of inappropriate antidiuretic hormone secretion (SIADH) [16]. Cuesta et al. confirmed this observation in only the of the patients [28]. Although the mechanisms are not precise, tachypnea was independently associated with hyponatremia as well as hypernatremia, in which tachypnea contributed to insensitive body fluid loss. The fluid loss was further exacerbated by reduced overall oral intake, which worsened the situation [29]. As mentioned in Cuesta’s study, Khann et al. reported that hyponatremia could not be totally explained with SIADH, and other factors could play a role in patients with COVID-19 infection [16,30]. Yousaf et al. defined many factors contributing to SIADH by affecting the activation of secondary pathways of ADH, such as interleukin-6 [31].

Another study found that, similar to our study, patients with hypernatremia exhibited significantly higher 30-day mortality and longer lengths of stay when compared to normonatremic patients [13]. In their study, Liu et al. found a significant difference in the presence of hypernatremia and hyponatremia between patients with and without COVID-19, even associating hypernatremia with a higher risk of mortality [15].

Hypernatremia was observed in patients requiring intensive care with SARS-CoV-2 pneumonia. This hypernatremia, seen in seriously ill patients, was resistant to treatment. There was no correlation between plasma sodium concentrations and sodium intake in these patients. These findings may be explained by abnormally increased renal sodium reuptake caused by increased angiotensin II activity, possibly due to severe SARS-CoV-2-induced downregulation of angiotensin-converting enzyme 2 (ACE2) receptors. We believe that hypernatremia in intensive care patients with SARS-CoV-2 pneumonia exacerbates the severity of the disease, leading to an increased need for invasive mechanical ventilation, inotropic support, and mortality.

The limitation of the study mainly could be stated that the patients’ type of hyponatremia and hypernatremia was unknown, as the exact volume given to the patients was not stated, and the volume status of the patients was not present in the study design. Dysnatrmeia, similarly, could not be entirely excluded, as the glucose status of the patients was present in the same sampling type. However, a repeated sampling of glucose was not performed for confirmation.

## 5. Conclusions

Our study validates that hypernatremia is an important risk factor in ICU patients hospitalized for SARS-CoV-2 infection for IMV, inotropic support, and mortality. Comorbidities, such as diabetes mellitus and renal failure associated with hypernatremia, exacerbate the disease course.

This complex relationship underlies the importance of proper electrolyte management, especially in patients who were under severe stress and organ failure.

## Figures and Tables

**Table 1 medicina-60-01019-t001:** Demographic parameters, comorbidities, and treatment modalities.

Demographic Parameters and Treatment Duration		No of Patients (*n* = 209)
**Gender**	Male (%)	114 (54.5)
	Female (%)	95 (45.5)
Age (years, SD)		68.15 (±13.81)
Body Mass Index (SD)		26.66 (±2.66)
RT-PCR Positivity to Admission (Days, 25th–75th)		5 (2–10)
Mechanical Ventilation Duration (Days, 25th–75th)		1 (0–6)
Non-invasive Mechanical Ventilation Duration (Days, 25th–75th)		2 (1–5)
LOS ICU (Days, 25th–75th)		6 (2–12)
Total Hospitalization Duration (Days, 25th–75th)		11 (5–18)
Culture Positivity (%)		78 (37.3)
**Comorbidities**		
Diabetes Mellitus		66 (31.6)
Hypertension		86 (41.1)
Coronary Arterial Disease		24 (11.5)
Congestive Heart Failure		14 (6.7)
Pulmonary Thromboembolism		10 (4.8)
Cerebrovascular Event		7 (3.3)
Chronic Renal Failure		3 (1.4)
**Nutritional and Respiratory Support**		
Total Parenteral Support Requirement		20 (9.6)
Enteral Support Requirement		110 (52.6)
**Treatment Modalities and Overall Mortality**		
Favipiravir		102 (48.8)
Tocilizumab		18 (8.6)
Nephrotoxic Antibiotic Therapy		47 (22.5)
Inotropic Support Requirement		124 (59.3)
Intravenous Glucocorticoid Requirement (Low)		81 (38.8)
Intravenous Glucocorticoid Requirement (High)		112 (53.6)
Mortality in ICU		124 (59.3)

**SD:** Standard deviation, **RT-PCR:** Reverse transcription polymerase chain reaction. The definition of non-invasive mechanical ventilation includes high-flow oxygenation requirements and continuous and bi-level pressure-support ventilation modes. The definition of culture positivity includes any positive bacterial and viral result taken from a patient upon intensive care unit admission, regardless of sampling origin. Any glucocorticoid regimen that includes a dosage of more than 1 mg/kg equivalent of methylprednisolone was defined as a high-dosage regimen.

**Table 2 medicina-60-01019-t002:** Comparison between laboratory parameters upon admission and last evaluation.

Paired Samples *T*-Test	Sampling Time	Mean	SD	*p*
Sodium (mEq/L)	Admission	139.44	6.80	**0.001**
	Last Evaluation	142.41	8.69	
Hemoglobin (g/dL)	Admission	13.03	2.81	**0.001**
	Last Evaluation	12.05	2.92	
White Blood Cell (10^9^/L)	Admission	13.33	7.09	**0.002**
	Last Evaluation	15.06	7.86	
Lymphocyte (10^9^/L)	Admission	0.96	1.71	**0.001**
	Last Evaluation	1.25	1.91	
Neutrophil (10^9^/L)	Admission	12.51	9.19	0.221
	Last Evaluation	13.38	7.39	
Platelets (10^9^/L)	Admission	258.23	119.17	**0.001**
	Last Evaluation	218.51	128.36	
Ferritin (ng/mL)	Admission	844.93	569.92	0.131
	Last Evaluation	898.79	588.70	
D-Dimer (mg/L)	Admission	8.793	24.663	0.526
	Last Evaluation	7.680	11.169	
Procalcitonin (ng/mL)	Admission	3.8	13.5	0.444
	Last Evaluation	4.8	15.6	
Creatinine Kinase (U/L)	Admission	196.07	287.86	0.155
	Last Evaluation	349.61	1410.65	
LDH (U/L)	Admission	612.72	677.78	**0.010**
	Last Evaluation	877.96	1496.24	
Glomerular Filtration Rate	Admission	70.48	29.25	0.261
	Last Evaluation	68.33	35.19	
Creatinine (mg/dL)	Admission	1.17	0.83	**0.001**
	Last Evaluation	1.43	1.27	
Potassium (mEq/L)	Admission	4.34	0.71	0.483
	Last Evaluation	4.29	0.99	
AST (U/L)	Admission	141.68	671.52	**0.041**
	Last Evaluation	308.71	943.93	
ALT (U/L)	Admission	109.11	567.28	**0.038**
	Last Evaluation	189.24	641.40	
C-Reactive Protein (mg/L)	Admission	127.22	86.89	**0.001**
	Last Evaluation	92.30	82.57	
Albumin (g/L)	Admission	28.30	5.23	0.112
	Last Evaluation	26.74	13.09	
ESR (mm/h)	Admission	60.44	29.04	**0.001**
	Last Evaluation	50.47	33.00	

*p* values of 0.05 and below were accepted as statistically significant.

**Table 3 medicina-60-01019-t003:** Correlation between admission sodium status, treatment modalities, and comorbidities.

	Spearman Correlation and *p*-Value	Admission Sodium Evaluation		
		Hypernatremia	Normal	Hyponatremia
Invasive Mechanical Ventilation Duration	Correlation	**0.171**	−0.087	−0.044
	*p*-value	**0.014**	0.211	0.523
Diabetes Mellitus	Correlation	**0.144**	−0.121	0.019
	*p*-value	**0.037**	0.080	0.786
Hypertension	Correlation	0.086	**−0.139**	0.090
	*p*-value	0.214	**0.044**	0.194
Chronic Renal Failure	Correlation	**0.184**	**−0.168**	0.040
	*p*-value	**0.008**	**0.015**	0.567
Enteral Support Requirement	Correlation	**0.242**	−0.114	−0.074
	*p*-value	**0.000**	0.100	0.285
Inotropic Support Requirement	Correlation	**0.160**	−0.072	−0.053
	*p*-value	**0.021**	0.302	0.449
Mortality in ICU	Correlation	**0.163**	−0.080	−0.047
	*p*-value	**0.018**	0.251	0.503

The definition of non-invasive mechanical ventilation includes high-flow oxygenation requirements and continuous and bi-level pressure-support ventilation modes. *p* values of 0.05 and below were accepted as statistically significant.

**Table 4 medicina-60-01019-t004:** Binominal regression analysis between hypernatremia and mortality.

Mortality in ICU	B	SE	Wald	Odds Ratio	*p*-Value
Constant	−7.213	5.885	1.502		
Age	0.052	0.026	3.959	1.053	**0.047**
Body Mass Index	0.211	0.093	5.214	1.235	**0.022**
Gender	−1.375	0.643	4.581	0.253	**0.032**
Total Parenteral Support Requirement	2.222	1.292	2.960	9.226	0.085
Enteral Support Requirement	2.293	0.694	10.923	9.906	**0.001**
MV Duration	0.135	0.233	0.334	1.144	0.563
NIMV Duration	0.040	0.236	0.028	1.040	0.867
Intensive Care Admission Duration	−0.128	0.229	0.311	0.880	0.577
Diabetes Mellitus	0.168	0.717	0.055	1.183	0.815
Hypertension	0.036	0.608	0.003	1.036	0.953
Coronary Arterial Disease	1.312	0.897	2.142	3.714	0.143
Congestive Heart Failure	0.354	1.061	0.111	1.424	0.739
Pulmonary Thromboembolism	0.280	1.431	0.038	1.322	0.845
Cerebrovascular Event	−2.934	1.976	2.204	0.053	0.138
Glucocorticoid Requirement (Low)	−2.080	1.230	2.860	0.125	0.091
Glucocorticoid Requirement (High)	−1.454	1.231	1.395	0.234	0.238
Favipiravir	0.382	0.581	0.431	1.465	0.512
Hypernatremia	0.882	0.923	0.914	2.416	0.339
Hemoglobin	−0.129	0.092	1.986	0.879	0.159
White Blood Cell	−0.020	0.051	0.160	0.980	0.689
Platelets	−0.001	0.003	0.275	0.999	0.600
Ferritin	0.001	0.001	4.746	1.001	**0.029**
D-Dimer	0.031	0.021	2.175	1.032	0.140
Procalcitonin	−0.002	0.025	0.006	0.998	0.938
C-Reactive Protein	0.009	0.004	6.422	1.009	**0.011**
ESR	−0.011	0.010	1.202	0.989	0.273
AST	−0.007	0.004	2.248	0.993	0.134
ALT	0.010	0.008	1.588	1.011	0.208

**SE:** Standard Error, **MV:** Mechanical Ventilation, **NIMV:** Non-invasive mechanical ventilation. **ESR:** Erythrocyte sedimentation rate, **AST:** Aspartate aminotransferase, **ALT:** Alanine transaminase.

## Data Availability

All relevant data are within the paper.

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
