# Peer review of "Dysnatremia as a Mortality Marker in Intensive Care Patients with SARS-CoV-2 Infection: A Retrospective Study"

_medicina, 2024, doi:10.3390/medicina60071019_

Round 1

Reviewer 1 Report

Comments and Suggestions for Authors

The paper looks good but the following improvements should be considered:

1- The introduction: I don't see the part that you mentioned about inflammatory biomarkers and their association between hyponatremia relevant to the rationale of this study.

You should also focus on how hyponatremia might affect severity. You should actually define what's severe COVID-19.

2-Methods: The definition of a severe COVID-19 is not clear. Which patient outcomes will be detected? Patient's are already having a severe disease since they are hospitalized. Is this your outcome? No mortality?

Moreover, can you compare the current findings to non-COVID-19?

3- The results are too confusing and you only need to present the significant results and reduce the number of tables.

4- The discussing lacks explanation of the results and only focuses on comparing the results to previous research findings.

5-The conclusion says that hypernatremia is a significant risk factor. For what? It's not clear again.

Overall, the rationale is not clear and needs to be elaborated more. Are we studying the effect of hypernatremia on COVID-19 outcomes? OR impact of COVID-19 on patients with hypernatremia? Why no controls were used? What about the underlying comorbidities of the patients since they were admitted to the ICU and might have other underlying risk factors than COVID-19.

Comments on the Quality of English Language

The manuscript might benefit from additional proofreading.

Author Response

For research article

Response to Reviewer X Comments

1. Summary

We would like to thank the editor and reviewers for the time you spent on this manuscript, and your contributions to science. Please find the detailed responses below and the corresponding revisions/corrections highlighted/in track changes in the re-submitted files.

2. Questions for General Evaluation

Reviewer’s Evaluation

Response and Revisions

Does the introduction provide sufficient background and include all relevant references?

Yes/Can be improved/Must be improved/Not applicable

We revised our manuscript in line with the suggestions of reviewers, and its language was also reviewed again.

Are all the cited references relevant to the research?

Yes/Can be improved/Must be improved/Not applicable

Is the research design appropriate?

Yes/Can be improved/Must be improved/Not applicable

Are the methods adequately described?

Yes/Can be improved/Must be improved/Not applicable

Are the results clearly presented?

Yes/Can be improved/Must be improved/Not applicable

Are the conclusions supported by the results?

Yes/Can be improved/Must be improved/Not applicable

3. Point-by-point response to Comments and Suggestions for Authors

REVIEWER 1:

Comments 1:  The paper looks good but the following improvements should be considered:

The introduction: I don't see the part that you mentioned about inflammatory biomarkers and their association between hyponatremia relevant to the rationale of this study.

Response 1:

HOPE is an international study registry of 4664 hospitalized SARS-CoV-2 infection patients. In this study, 20.5% of patients were reported to have hyponatremia, and hypernatremia was observed in 3.7% of patients 8. Proinflammatory cytokines such as IL-1b and IL-6 are known to stimulate hypothalamic arginine vasopressin secretion 9-11. Supporting this, the study by Berni et al. reported that IL-6 levels were inversely proportional to serum sodium levels. The coexistence of hyponatremia and elevation of IL-6 levels has been shown in SARS-CoV2 patients 12. Therefore, it can be predicted that dysnatremia affects prognosis and may be associated with mortality in patients admitted to SARS-CoV2 intensive care with SARS-CoV-2.

We agree with this comment. Therefore, we have change paragraph to;

HOPE is an international study registry of 4664 hospitalized patients with SARS-CoV-2 infection. In this study, 20.5% of patients were reported to have hyponatremia, and hypernatremia was observed in 3.7% of patients 16. It has been reported in many publications that hyponatremia may be due not only to hypovolemia, but also to some hypothalamic arginine vasopressin secretion stimulation 17-20.

You should also focus on how hyponatremia might affect severity. You should actually define what's severe COVID-19.

Changes in plasma sodium concentration are determined by changes in water balance, independent of the total body sodium amount. The two basic mechanisms responsible for regulating water metabolism are the antidiuretic hormone and the feeling of thirst 5. While SARS-CoV-2 infection can cause multiple organ failure, it can also cause dysfunction of the renin-angiotensin-aldosterone system and disrupt water hemostasis with thirst and appetite abnormalities 6. Various electrolyte disorders were observed in SARS-CoV2 patients, but hypo-hypernatremia was particularly striking. Severe hyponatremia alone may be the leading cause of death or cause of permanent neurological changes. The most important risk factors for death in patients presenting with hyponatremia were found to be hypoxia and sepsis 7.

Thank you for pointing this out. We change to;

Changes in plasma sodium concentration are determined by changes in water balance, independent of the total body sodium amount. The two basic mechanisms responsible for regulating water metabolism are the antidiuretic hormone and the feeling of thirst 7. While SARS-CoV-2 infection can cause multiple organ failure, it has also been reported to cause dysfunction of the renin-angiotensin-aldosterone system, disrupting water homeostasis with thirst and appetite abnormalities. Hyponatremia occurs due to the effect of antidiuretic hormone (ADH). Symptoms are typically dependent on the rapidity of onset of the clinical condition. Neurological symptoms are possible in severe hyponatremia (serum sodium < 120 mEq/L). Hypernatremia is usually caused by either a deficit of total body water or by an inappropriately high sodium input.  In general, however, even during infusion of large amounts of sodium-containing solutions (as during treatment of acute hypovolemia), hypernatremia is infrequently observed and less pronounced.

Comments 2:  Methods: The definition of a severe COVID-19 is not clear. Which patient outcomes will be detected? Patient's are already having a severe disease since they are hospitalized. Is this your outcome? No mortality?

Moreover, can you compare the current findings to non-COVID-19?

Response 2:

Patients who have had SARS-CoV-2 pneumonia but are being treated in the hospital ward, and whose condition worsens with increasing respiratory distress requiring high-flow oxygen, NIMV, and IMV, are being admitted to the intensive care unit. We define severe those patients in the intensive care unit with SARS-CoV-2 pneumonia who require IMV, receive inotropic support, and have mortality.

                             We couln’t compare the findings to non-COVID-19 because we do not have data             available for that group.

Comments 3:  The results are too confusing and you only need to present the significant results and reduce the number of tables.

Response 3: Thank you for pointing this out.

As no correlation was present with hyponatremia, further regression analysis was not performed.

Regarding table count, the initial three tables consisted of demographic data and comparisons between and within groups. As such, we could not remove them. Table 4-5-6, on the other hand, are regression analysis performed for further investigation and if you deem it fit, we can omit them and/or present them as a supplementary. Some changes were made based on your suggestions

Comments 4:  The discussing lacks explanation of the results and only focuses on comparing the results to previous research findings.

Response 4: Thank you for pointing this out.

The discussion section has been changed according to your suggestions

Comments 5:  The conclusion says that hypernatremia is a significant risk factor. For what? It's not clear again. Overall, the rationale is not clear and needs to be elaborated more. Are we studying the effect of hypernatremia on COVID-19 outcomes? OR impact of COVID-19 on patients with hypernatremia? Why no controls were used? What about the underlying comorbidities of the patients since they were admitted to the ICU and might have other underlying risk factors than COVID-19.

Response 5: Thank you for pointing this out. We change the conclusion paragraph.

Our study had validated that hypernatremia was an important risk factor in ICU patients hospitalized for SARS-CoV-2 infection for IMV, inotropic support and mortality. Comorbidities such as diabetes mellitus and renal failure associate with hypernatremia exacerbate the disease course.

This complex relationship underlies the importance of proper electrolyte management, especially in patients who were under severe stress and organ failure.

4. Response to Comments on the Quality of English Language

Point 1:

Response 1: The language has been reviewed and corrected accordingly

5. Additional clarifications

Based on your suggestions, we expanded the manuscript by increasing the number of references.

Reviewer 2 Report

Comments and Suggestions for Authors

An important retrospective study highlighting dysnatraemia as a mortality

marker in ICU patients with SARS-CoV-2 infection. A large number of parameters have been studied to provide a deep insight on the subject matter for the readers. However, authors are advised to incorporate the suggested modifications.   

 Title:

 As the study is retrospective, therefore the title may be changed to

  Dysnatraemia as a Mortality Marker in Intensive Care Patients with SARS-CoV-2 Infection- A Retrospective Study.

Abstract: Line 50

 Loosing water through urine and feces, perspiration, and respiration is a physiological process, therefore the word water hemostasis  should be replaced with water homeostasis, a balanced and sensitive network of physiological controls necessary to maintain water levels. 

 Table-1:  Line -151

 Culture positivity recorded was 37.3%. The authors should also mention the predominance of microorganism if the information is available. This will add value to the manuscript.

 Table-2: Line 174

 Data have been nicely presented and provide deep insight about such changes. Sedimentation here speaks about Erythrocytes Sedimentation Rate (ESR) or something else? Please write for better understanding of the readers. Also, globulins concentration will provide a better picture to depict immunity of the patients. Therefore, if the data on globulins are available, may kindly be incorporated.

 Table-3: Line-187

 Mortality in ICU shows a negative correlation with normal and hyponatremia patients, whereas, the hyponatremia has been found an important risk factor in ICU patients hospitalized for SARS-CoV-2 infection. This may kindly be elucidated.

 Discussion:

The study initially in the abstract highlights dysfunction of the renin-angiotensin-aldosterone system (RAAS) in SARS COV-2 patients. But none of the components of RAAS viz., renin, angiotensinogen and aldosterone has neither been study nor discussed in the manuscript. The authors are suggested to modify the contents in the abstract.   

 Reference Section

  Each Reference number has been typewritten twice like 1.   1.       2.  2.  The same may be corrected

Corrections have made in the manuscript at 3-4 places. The same is attached. 

Author Response

For research article

Response to Reviewer X Comments

1. Summary

We would like to thank the editor and reviewers for the time you spent on this manuscript, and your contributions to science. Please find the detailed responses below and the corresponding revisions/corrections highlighted/in track changes in the re-submitted files.

2. Questions for General Evaluation

Reviewer’s Evaluation

Response and Revisions

Does the introduction provide sufficient background and include all relevant references?

Yes/Can be improved/Must be improved/Not applicable

We revised our manuscript in line with the suggestions of reviewers, and its language was also reviewed again.

Are all the cited references relevant to the research?

Yes/Can be improved/Must be improved/Not applicable

Is the research design appropriate?

Yes/Can be improved/Must be improved/Not applicable

Are the methods adequately described?

Yes/Can be improved/Must be improved/Not applicable

Are the results clearly presented?

Yes/Can be improved/Must be improved/Not applicable

Are the conclusions supported by the results?

Yes/Can be improved/Must be improved/Not applicable

3. Point-by-point response to Comments and Suggestions for Authors

REVIEWER 2:

Comments 1:  An important retrospective study highlighting dysnatraemia as a mortality marker in ICU patients with SARS-CoV-2 infection. A large number of parameters have been studied to provide a deep insight on the subject matter for the readers. However, authors are advised to incorporate the suggested modifications.   

 Title:

 As the study is retrospective, therefore the title may be changed to

  Dysnatraemia as a Mortality Marker in Intensive Care Patients with SARS-CoV-2 Infection- A Retrospective Study.

Response 1: Thank you for pointing this out. We  change the title as you suggested.

Comments 2: 

Abstract: Line 50

 Loosing water through urine and feces, perspiration, and respiration is a physiological process, therefore the word water hemostasis  should be replaced with water homeostasis, a balanced and sensitive network of physiological controls necessary to maintain water levels. 

Response 2: We agree with this comment. We changed the word in the manuscript.

 Comments 3:  Table-1:  Line -151

 Culture positivity recorded was 37.3%. The authors should also mention the predominance of microorganism if the information is available. This will add value to the manuscript.

Response 3: Thank you for pointing this out. We add this information.

78 patients (37.3%) showed culture positivity regarding additional bacterial involvement. In the culture of patients, the predominant organism was Staphylococcus aureus (n:31), followed by Candida (both albicans and non-albicans)(n:24), Enterococcus, Escherichia coli, Acinetobacter, Klebsiella pneumonia, and Pseudomonas aeruginosa. In some patients, there was also polymicrobial growth present.

 Comments 4:  Table-2: Line 174

 Data have been nicely presented and provide deep insight about such changes. Sedimentation here speaks about Erythrocytes Sedimentation Rate (ESR) or something else? Please write for better understanding of the readers. Also, globulins concentration will provide a better picture to depict immunity of the patients. Therefore, if the data on globulins are available, may kindly be incorporated.

Response 4: Thank you for pointing this out. We couln’t get the globulin values so we couln’t use. We change to ESR as you suggested.

 Comments 5:  Table-3: Line-187

 Mortality in ICU shows a negative correlation with normal and hyponatremia patients, whereas, the hyponatremia has been found an important risk factor in ICU patients hospitalized for SARS-CoV-2 infection. This may kindly be elucidated.

Response 5: We agree with this comment. Therefore, we emphasized this point by highlighting it in the text.

Although we predicted dysnatremia in our study, our results did not observe hyponatremia as a significant risk factor.

Comments 6:   Discussion: The study initially in the abstract highlights dysfunction of the renin-angiotensin-aldosterone system (RAAS) in SARS COV-2 patients. But none of the components of RAAS viz., renin, angiotensinogen and aldosterone has neither been study nor discussed in the manuscript. The authors are suggested to modify the contents in the abstract.

Response 6: Thank you for pointing this out. We change the paragraph to:

Introduction: The Severe Acute Respiratory Syndrome Coronavirus 2 (SARS-CoV-2) infection may cause acute respiratory failure but also remains responsible for many other pathologies, including electrolyte disorders. SARS-CoV-2 infection causes disorders in many systems and can disrupt water homeostasis with thirst and appetite abnormalities.

Comments 7:   Reference Section:  Each Reference number has been typewritten twice like 1.   1.       2.  2.  The same may be corrected

Response 7: We  changed according to your suggestions

4. Response to Comments on the Quality of English Language

Point 1:

Response 1: The language has been reviewed and corrected accordingly

5. Additional clarifications

Based on your suggestions, we expanded the manuscript by increasing the number of references.

Reviewer 3 Report

Comments and Suggestions for Authors

The authors have done a great deal of work, regarding patients with coronavirus respiratory infection and, mainly, its correlation with hypernatremia. However, I have the feeling that the results are presented in a rather confusing way, especially in the discussion section. The authors should state more clearly for the sake of their paper, the exact correlation of hypernatremia with the basic parameters they have concluded to, this might be also done in a seperate sentence/ paragraph. 

Otherwise, I believe the apper deserves publication after a proper linguistic control. 

Comments on the Quality of English Language

Language check needed. 

Author Response

For research article

Response to Reviewer X Comments

1. Summary

We would like to thank the editor and reviewers for the time you spent on this manuscript, and your contributions to science. Please find the detailed responses below and the corresponding revisions/corrections highlighted/in track changes in the re-submitted files.

2. Questions for General Evaluation

Reviewer’s Evaluation

Response and Revisions

Does the introduction provide sufficient background and include all relevant references?

Yes/Can be improved/Must be improved/Not applicable

We revised our manuscript in line with the suggestions of reviewers, and its language was also reviewed again.

Are all the cited references relevant to the research?

Yes/Can be improved/Must be improved/Not applicable

Is the research design appropriate?

Yes/Can be improved/Must be improved/Not applicable

Are the methods adequately described?

Yes/Can be improved/Must be improved/Not applicable

Are the results clearly presented?

Yes/Can be improved/Must be improved/Not applicable

Are the conclusions supported by the results?

Yes/Can be improved/Must be improved/Not applicable

3. Point-by-point response to Comments and Suggestions for Authors

REVIEWER 3:

Comments 1: The authors have done a great deal of work, regarding patients with coronavirus respiratory infection and, mainly, its correlation with hypernatremia. However, I have the feeling that the results are presented in a rather confusing way, especially in the discussion section. The authors should state more clearly for the sake of their paper, the exact correlation of hypernatremia with the basic parameters they have concluded to, this might be also done in a seperate sentence/ paragraph. Otherwise, I believe the apper deserves publication after a proper linguistic control. . 

Response 1: Thank you for pointing this out. We change the paragraph to:

This study demonstrated the association of hypernatremia in SARS-CoV-2 enfection patients in ICU with IMV, requirement for inotropic support, and enteral nutrition. Another finding of the study, the association of hypernatremia with comorbidities such as diabetes mellitus and renal failure, suggests that these conditions exacerbate the disease course.

4. Response to Comments on the Quality of English Language

Point 1:

Response 1: The language has been reviewed and corrected accordingly

5. Additional clarifications

Based on your suggestions, we expanded the manuscript by increasing the number of references.

Round 2

Reviewer 1 Report

Comments and Suggestions for Authors

The manuscript looks fine now, except for the results and I still believe that moving some of the information within the tables to the supplementary would be better for a clear visual representation of the significant findings.

Author Response

First of all, thank you for your suggestions to make our article more valuable.

We have shortened and reduced the tables as suggested by the reviewer.  We have also added the table of some of our findings as suplemantary materials. We think that the manuscript is appropriate in this form.
